# A Comprehensive Review on Bioprinted Graphene-Based Material (GBM)-Enhanced Scaffolds for Nerve Guidance Conduits

**DOI:** 10.3390/biomimetics10040213

**Published:** 2025-03-31

**Authors:** Siheng Su, Jilong Wang

**Affiliations:** 1Key Laboratory of Clean Dyeing and Finishing Technology of Zhejiang Province, College of Textile and Garment, Shaoxing University, Shaoxing 312000, China; 2Shaoxing Sub-Center of National Engineering Research Center for Fiber-Based Composites, Shaoxing University, Shaoxing 312000, China; 3Shaoxing Key Laboratory of High Performance Fibers & Products, Shaoxing University, Shaoxing 312000, China; 4Department of Mechanical Engineering, California State University, Fullerton, CA 92831, USA

**Keywords:** graphene-based materials, nerve guidance conduits, 3D printing, tissue engineering

## Abstract

Peripheral nerve injuries (PNIs) pose significant challenges to recovery, often resulting in impaired function and quality of life. To address these challenges, nerve guidance conduits (NGCs) are being developed as effective strategies to promote nerve regeneration by providing a supportive framework that guides axonal growth and facilitates reconnection of severed nerves. Among the materials being explored, graphene-based materials (GBMs) have emerged as promising candidates due to their unique properties. Their unique properties—such as high mechanical strength, excellent electrical conductivity, and favorable biocompatibility—make them ideal for applications in nerve repair. The integration of 3D printing technologies further enhances the development of GBM-based NGCs, enabling the creation of scaffolds with complex architectures and precise topographical cues that closely mimic the natural neural environment. This customization significantly increases the potential for successful nerve repair. This review offers a comprehensive overview of properties of GBMs, the principles of 3D printing, and key design strategies for 3D-printed NGCs. Additionally, it discusses future perspectives and research directions that could advance the application of 3D-printed GBMs in nerve regeneration therapies.

## 1. Introduction

The nervous system is one of the most complex parts of the human body, comprising the central nervous system (CNS) and the peripheral nervous system (PNS). The CNS, which includes the brain and spinal cord, serves as the command center of the nervous system, regulating human behavior through its interactions with the PNS, which consists of the nerves located outside the brain and spinal cord. Consequently, injuries to the PNS can lead to significant issues, including loss of muscle control, sensory degeneration, and even paralysis, as these injuries hinder communication with the CNS. Every year, more than 1 million people suffer from PNS injuries [1], which can arise from various causes such as trauma, fractures, repetitive stress, and pre-existing medical conditions. Currently, autologous transplantation is considered the gold standard for treating severe peripheral nerve injuries (PNIs), although this surgical procedure presents several challenges. While advancements in microsurgical techniques have improved outcomes, autologous grafting is typically effective only for nerve gaps measuring 5 cm or less [2]. Additionally, it is limited by the availability of donors and carries risks of infections and other complications. Alternatively, donor-free tissue engineering approaches, such as nerve conduits, are being explored as viable solutions for managing severe PNS injuries. Nerve conduits function as bridging devices that encapsulate the proximal and distal segments of the nerve, guiding axonal growth along the original pathway while minimizing issues related to the midsection [3]. When selecting appropriate materials for nerve conduit development, several criteria must be taken into account, including material properties and biological responses. The key factors include: 1. Biocompatibility [4]: This is the most critical criterion for materials intended for use in the human body. Any material employed must be non-toxic and biocompatible to prevent severe side effects; 2. Biodegradability [5]: Materials should degrade at a suitable rate to eliminate the need for secondary surgeries; 3. Appropriate mechanical properties [6]: Suitable mechanical properties are essential to support the nerve structure, avoiding stress concentration at the injury site and preventing collapse during normal movement; 4. Appropriate porosity [7]: A porous structure is vital for mimicking the extracellular matrix (ECM) of natural tissues, facilitating the transport of water, nutrients, and cells; 5. Neural transmission [8]: In the neural system, synaptic actions generate neural transmission, which is essential for neuronal communication. Therefore, electrically conductive materials are required in nerve conduits to promote neurite growth and neural regeneration.

Given these criteria, graphene-based materials (GBMs) hold considerable promise for use in nerve conduits. GBMs refer to a class of materials associated with graphene, a two-dimensional sp^2^ carbon-based nanomaterial that was first isolated and characterized in 2004 [9]. Due to its high surface area, exceptional macromolecule adsorption capacity, low toxicity, and excellent thermal and electrical conductivities [10,11,12], graphene has garnered substantial attention from both research and industry sectors, leading to its application in electronics, thermal management, and biomedical fields. For example, a peptide nucleic acid-functionalized graphene biosensor was developed to detect the prostate cancer biomarker hs-miR-141, achieving high specificity and a detection limit of 0.6 Am [13]. Furthermore, graphene has been utilized in chemotherapy-enhanced photothermal therapy for cancer treatment, thanks to its high photo-to-thermal conversion efficiency and extensive surface area, which allows for significant drug loading capacity [14,15]. In addition to biosensors and therapeutic applications, studies indicate that graphene can promote cell proliferation and differentiation by enhancing the surface bioactivation of materials when coated on polymeric scaffolds [16], thereby accelerating tissue regeneration [17]. Graphene oxide (GO) was immobilized on electrospun poly(caprolactone) (PCL) nanofibers to encourage skeletal muscle cell growth and differentiation for tissue engineering purposes [18]. Moreover, graphene derivatives can form crosslinks through covalent and non-covalent bonds with acrylic acid and similar monomers, resulting in ultra-strong and tough hydrogels with a tensile strength of 248 kPa and toughness of 7124 kJ/m^3^ [19], providing adequate support for tissue conduits. The pore size of graphene-interconnected polymeric hydrogels can also be adjusted by modifying the wettability and concentration of GO [20], enabling selective transport of cells and nutrients in tissue engineering. Additionally, a highly conductive hydrogel was developed by reducing self-assembled GO-coated agarose microbeads into three-dimensionally connected rGO hydrogels via thermal annealing [21]. These characteristics indicate that GBMs represent an excellent candidate for nerve conduits, compared to other materials like synthetic polymeric materials, including polylactic acid (PLA) [22], polycaprolactone (PCL) [23], and polyurethane (PU) [24]; natural polymeric materials, including gelatin [25], alginate [26], and collagen [27]; and conductive polymeric materials, including polypyrrole (PPy) [28] and their composites [29].

Traditional methods for producing tissue conduits, such as solvent casting [29], micropatterning [30], and electrospinning [31], offer controllable porosity and morphology but require complex equipment. In contrast, additive manufacturing, or 3D printing, presents new opportunities for the production of nerve conduits. This technology enables the creation of precise shapes based on three-dimensional models and offers exceptional control over the internal design of printed structures with high resolution. Three-dimensional bioprinting can be categorized into light-based, inkjet-based, and extrusion-based techniques [32], providing a wide range of material options for nerve conduit fabrication.

While numerous reviews cover related topics, many focus either on GBMs as NGCs through various manufacturing methods [33,34,35] or on 3D printing techniques for NGCs using different materials [36,37,38]. However, to the best of our knowledge, few have specifically explored the production of GBM-based NGCs using 3D printing methods in recent years. This review offers a comprehensive examination of 3D-printed GBM-based NGCs. We first examine the mechanical, electrical, and cytotoxic properties of GBMs to establish their suitability for nerve conduits. Next, we discuss various additive manufacturing processes applicable to biomedical applications. Following this, we explore 3D-printed GBM-based nerve conduits for treating PNS injuries. Finally, we highlight research perspectives on 3D-bioprinted GBM-enhanced scaffolds for nerve conduits.

## 2. Graphene-Based Materials (GBMs)

### 2.1. Classification

Graphene consists of a single layer of carbon atoms arranged in a honeycomb lattice and was first termed by Mouras et al. in 1987 [39] to describe the fundamental building block of graphite. Although the structure of graphene has been examined using electron microscopy since the 1960s, it was not until 2004 that mono-layer graphene was successfully isolated from graphite and transferred onto a silicon wafer using the Scotch Tape Technique by Novoselov and Geim [40]. Since this groundbreaking achievement, graphene has generated significant interest and excitement among researchers and engineers.

GBMs can be classified based on either the number of layers or the degree of oxidation. In terms of layer count, graphene is categorized as single-layer, bi-layer, few-layer (2 to 10 layers), or multi-layer (more than 10 layers). Additionally, graphene derivatives are distinguished by their oxidation levels: pristine graphene (PG), graphene oxide (GO), and reduced graphene oxide (rGO) [41]. PG is produced through methods like mechanical cleavage and chemical vapor deposition (CVD) from graphite. In contrast, GO consists of oxidized graphene sheets that contain epoxy, hydroxyl, and carboxyl functional groups, which are generated during the oxidation and exfoliation of graphite in top-down synthesis. While these functional groups facilitate the conjugation of other molecules for functionalization, they also decrease the material’s electrical conductivity [42]. rGO, as the name suggests, is obtained by reducing GO [43]. This reduction process can partially restore the electrical conductivity of rGO while preserving the advantageous dispersion characteristics of GO in various solvents.

### 2.2. Synthesis of Graphene

Generally, there are two methods to produce graphene: ‘top-down’ and ‘bottom-up’ methods (Figure 1) [44]. In the ‘top-down’ approach, mechanical or chemical techniques are employed to break down bulk materials like graphite into smaller graphene flakes, followed by purification to isolate mono-layer, few-layer, and multi-layer graphene. Conversely, the ‘bottom-up’ method involves growing carbon atoms onto a substrate using deposition techniques such as CVD.

#### 2.2.1. “Top-Down” Approach

Mechanical exfoliation from bulk materials uses mechanical forces to overcome the Van der Waals forces between graphene layers. The Scotch tape technique, for instance, involves applying normal force repeatedly to highly ordered pyrolytic graphite (HOPG), thinning it layer by layer until a single graphene layer remains. While this method can produce high-quality, large-area 2D graphene, it is time-consuming, inefficient, and not scalable for industrial production [46]. A sectioning technique inspired by the Scotch tape method was later developed to produce few-layer graphene from HOPG [47]. In this approach, graphene layers are cleaved and sliced by an ultrasharp single-crystal diamond wedge with ultrasonic oscillations and collected in a water bath. The technique successfully generated few-layer graphene with dimensions of 900 µm × 300 µm from a 1 mm × 0.5 mm HOPG sample.

Liquid-phase exfoliation (LPE) is another top-down method for producing graphene. In LPE, HOPG is dispersed in organic solvents like N-methyl-2-pyrrolidone (NMP) and N,N′-dimethylformamide (DMF) through sonication. This process reduces Van der Waals forces and increases the spacing between graphene layers, allowing the production of large quantities of high-quality graphene [48]. LPE involves three stages [49]: (1) High-intensity sonication breaks down large HOPG flakes, creating kink bands; (2) Graphite stripes are peeled off from HOPG with solvent assistance; (3) Edge tearing and intercalation of peeled graphite stripes produce few-layer graphene flakes. LPE can yield graphene with widths ranging from 20 nm to 1000 nm and thicknesses between 0.35 nm and 20 nm, depending on the raw material, solvent, and sonication technique used. In addition to mechanical exfoliation and LPE, methods such as electrochemical exfoliation and rapid thermal annealing have been developed for large-scale graphene production [50]. However, the oxidation, sonication, or electrochemical treatments in top-down approaches often introduce defects in the graphene structure [51]. On the other hand, bottom-up methods produce higher-quality graphene but require more complex equipment and controlled synthesis environments.

#### 2.2.2. “Bottom-Up” Approach

Several bottom-up techniques are available to produce high-quality graphene, including epitaxial growth, chemical vapor deposition (CVD), electric discharge, and various laser-related methods. Among these, CVD is particularly notable for producing large-scale, high-yield mono- and bi-layer graphene [44]. The CVD process involves mixing a homogeneous gas phase with a heterogeneous substrate under specific temperature and pressure conditions [52]. For instance, Duan et al. utilized a high H_2_/CH_4_ ratio precursor to grow bi-layer graphene on Cu foil in a low-pressure (5 mbar) and high-temperature (1050 °C) CVD process, achieving up to 99% graphene coverage on the substrate [53].

Traditionally, CVD produces graphene on metal substrates due to their catalytic properties, which requires further isolation or transfer to the target substrate. However, avoiding damage or contamination during this transfer process poses significant challenges. Two approaches have been developed to address this issue: direct growth of graphene on desired nonmetal solid substrates or on molten/flexible substrates [54]. For example, mono-layer graphene was successfully formed on two Si/SiOx substrates using a “sandwich” structure, with an Ar/H_2_/CH_4_ gaseous precursor at 1185 °C under ambient pressure. In this case, the oxides on the wafers acted catalytically [55]. Liu et al. [56] demonstrated the production of a 30 cm × 6 cm graphene sheet on a commercial soda-lime glass substrate using an Ar/H_2_/CH_4_ gaseous precursor. The glass substrate melted at the growth temperature of 1020 °C due to its low softening point of 620 °C. Flexible nonmetallic substrates have also been explored to avoid damage during the transfer process. For substrates with low melting points, such as hexagonal boron nitride (hBN), plasma-enhanced CVD (PECVD) can be employed to reduce the synthesis temperature and prevent substrate melting [57].

### 2.3. Properties of Graphene-Based Materials

Graphene-based materials offer significant potential for NGCs due to their remarkable properties, including ease of functionalization, excellent mechanical strength, high electrical conductivity, and superior biocompatibility [58]. This section provides a detailed analysis of these properties.

#### 2.3.1. Mechanical Properties

In PG, carbon atoms are arranged in a honeycomb lattice, with a C-C bond length of 1.42 Å. These atoms are covalently bonded through in-plane σ bonds formed by sp^2^ orbitals, with π orbitals partially occupying either side of the graphene sheet. For multi-layer graphene, the inter-layer spacing is approximately 3.34 Å, which helps maintain the material’s lamellar structure [59]. Unlike PG, GO contains some sp^3^-hybridized bonds due to the presence of functional groups, with the ratio of sp^2^ to sp^3^ hybridization being influenced by the synthesis and preparation methods [60]. In rGO, many functional groups are removed through reduction processes, leaving defects on the graphene sheet.

Mono-layer graphene is characterized by its brittleness, with an ultra-high elastic modulus of 1.0 TPa [61] and a fracture strength of 130 GPa [62] as measured by atomic force microscopy (AFM) using a diamond-coated tip. These properties arise from the exceptionally strong covalent bonds within its hexagonally arranged 2D structure. The incorporation of functional groups leads to the formation of ripples on the graphene sheet, which in turn deteriorates the mechanical properties of GO. Computer simulations indicate that the elastic modulus of GO with 40% epoxide coverage is 403 GPa, while that of GO with 40% hydroxyl coverage is 452.5 GPa [63]. Experimental findings reveal that the effective elastic modulus of a mono-layer GO membrane with 40% sp^3^-bonded is lower than the predicted values from simulations, measuring at 207.6 ± 23.4 GPa as determined by AFM. Additionally, the study found that two-layer and three-layer GO membranes exhibit elastic moduli of 444.8 ± 235.3 GPa and 665.5 ± 34.6 GPa, respectively [64]. These results suggest that the inter-layer bonding between GO sheets is sufficiently robust to prevent sliding. Although oxidation groups are removed from GO, defects present in rGO significantly affect its elastic modulus, leading to a much lower elastic modulus in comparison to PG. Mono-layer chemically rGO demonstrated an elastic modulus of 250 ± 150 GPa [65], which is similar to that of GO but considerably lower than that of PG.

#### 2.3.2. Electrical Properties

The electrical performance of GBMs is influenced by the synthesis methods employed. For mechanically exfoliated graphene, charge carrier mobility can reach up to $1\times 10^{6}\;{cm}^{2}V^{-1}s^{-1}$ at low temperatures, and $2\times 10^{5}\;{cm}^{2}V^{-1}s^{-1}$ at room temperature. In contrast, graphene synthesized via CVD exhibits a mobility ranging from $1\times 10^{4}\sim 5\times 10^{4}\;{cm}^{2}V^{-1}s^{-1}$ [66,67]. Chemical exfoliation leads to lower mobility due to the smaller overlapping size of flakes, resulting in a mobility of approximately $100\;{cm}^{2}V^{-1}s^{-1}$ [66]. The electrical conductivity of high-quality CVD-grown PG can reach values as high as $10^{4}\sim 10^{5}\;S\cdot {cm}^{-1}$ [68,69,70,71].

In contrast to graphene sheets, the presence of sp^3^ bonds and the disruption of sp^2^ bonds in GO result in a high electrical resistivity of $1.64\times 10^{4}\;\Omega \cdot m$, limiting its potential applications. One approach to enhance its electrical conductivity is through the reduction of GO to rGO [42,72,73,74,75]. The electrical properties of rGO are influenced by structural defects, functional groups, and layer disorder [76], all of which can be manipulated through reduction methods, reducing agents, and reduction duration. During the reduction process, carbon atoms from the GO sheet may be removed, creating structural defects such as ‘holes’, which can increase the electron travel path [42]. Moreover, low reduction efficiency can result in a high concentration of sp^3^ groups, which not only obstruct the electron travel path but also trap electrons.

For multi-layer rGO, the inter-layer structure has a lesser impact on electrical conductivity compared to defects and functional groups; however, a larger d-spacing can facilitate the restoration of sheet structure and enhance conductivity [77]. Consequently, rGO with fewer oxygen-containing groups and defects would exhibit improved electrical conductivity. Research has shown that chemically reduced GO (rGO) treated with hydroiodic acid achieves an electrical conductivity of 10,330 S/m [42], significantly surpassing that of GO, which is ($4.57\times 10^{-6}\;S/m$ [78]). Therefore, utilizing an appropriate reduction method that effectively removes oxygen functional groups while preserving structural integrity can enable the electrical mobility and conductivity of rGO to approach that of PG. For example, Hu et al. [75] utilized a 3000 K Joule heating method to produce rGO with an electrical mobility of $320\;{cm}^{2}V^{-1}s^{-1}$ and a conductivity of $6300\;S\cdot {cm}^{-1}$ at room temperature [75]. The process involved initially reducing GO at 1000 K in an inert gas environment with a slow heating rate to facilitate the release of oxygen functional groups in gaseous form. The temperature was then increased to 2000 K to eliminate any remaining functional groups, followed by a rapid rise to 3000 K to prevent significant structural alterations in the rGO film.

#### 2.3.3. Cytotoxicity Properties

Biocompatibility is a key consideration for materials intended for biomedical applications and is typically evaluated using both in vitro and in vivo experiments. However, the toxicity profile of GBMs remains inconsistent across studies. While some research highlights the ability of GBMs to enhance cell growth, others raise concerns about their potential health risks. This variation may be due to differences in the properties of synthesized graphene and the routes of exposure, such as inhalation, ingestion, dermal contact, etc. [79]. As shown in Table 1, various factors, including dosage, concentration, functionalization, synthesis methods, lateral size, number of layers, and morphology, significantly influence their cytotoxicity [80].

In vitro cytotoxicity studies are cost-effective and convenient, typically assessing parameters such as cell viability, reactive oxygen species (ROS) induction, lactate dehydrogenase (LDH) levels, and cell morphology. Research has demonstrated that the cytotoxicity of aqueous PG is both dosage- and cell line-dependent. At low dosage (<10 μg/mL), PG does not induce significant changes in cellular morphology or cause damage [81,82]. However, studies have shown that HDFs exhibit only 80% cell viability at 15.6 μg/mL after 24 h exposure to CVD-grown PG, likely due to membrane rupture and nuclear DNA damage [83]. Interestingly, PG shows no significant cytotoxicity in L-929 fibroblast cells, even at concentrations as high as 200 μg/mL [83]. Rather than inducing damage, graphene films coated on culture coverslips promote cell growth and accelerate tissue repair by activating epidermal growth factor receptors [82,84,85].

GO also exhibits dose-dependent cytotoxicity, primarily due to increased ROS levels that cause mitochondrial damage and apoptosis activation [85,86,87]. Additionally, chemically synthesized GO may contain metal residues, complicating cell interactions [85]. Studies suggest that mono-layer and smaller-sized GO show lower toxicity [88,89], whereas increased surface carbon radicals contribute to plasma membrane adsorption, lipid peroxidation, and membrane damage, ultimately leading to cell death [90]. The cytotoxicity mechanisms of rGO are similar to PG due to its hydrophobic nature, which promotes aggregation on cell surfaces and biomolecule adsorption, leading to cellular damage rather than ROS generation as seen with GO [91]. Furthermore, certain reducing agents used to produce rGO can result in irregular shapes with sharp protrusions, causing physical membrane stress, protein leakage, and increased cytotoxicity [87]. Therefore, proper preparation and functionalization are crucial for enhancing the stability and reducing the toxicity of GBMs.

The in vivo biocompatibility of GBMs is influenced by their physical and chemical properties, functionalization, dosage, and exposure duration. Although the mechanisms of in vivo biocompatibility are similar to those observed in vitro, the complexity and high cost of in vivo experiments have limited the number of studies in this area [92]. Most in vivo research on GBMs biocompatibility focuses on administration routes such as intravenous injection, inhalation, instillation, or subcutaneous injection, which are crucial for assessing toxicity. However, studies examining the toxicity of GBMs through implantation remain scarce.

Researchers have found that GBMs exhibit moderate biocompatibility when implanted in subcutaneous and peritoneal tissues. Zha et al. [93] investigated the effects of CVD-grown PG and GO foam for subcutaneous implantation in rat models. Over a two-week period, the study found no significant hematologic, hepatic, or renal toxicity, and neither material triggered noticeable inflammatory responses at the implant site or in the liver. In long-term studies (7 months), no behavioral or activity differences were observed between experimental and control animals. Langer et al. [94] further demonstrated that the oxidation state of GO significantly impacts its biocompatibility. Reduced GO was associated with quicker immune cell infiltration, uptake, and clearance from subcutaneous implant sites, without signs of inflammation or tissue damage. In contrast, GO with higher oxidation levels elicited a stronger pro-inflammatory response in peritoneal implantation. Additionally, several studies [95,96,97] suggest that functionalizing GBMs with polymers, such as PEG, improves their biodistribution and stability, enhancing overall biocompatibility.
biomimetics-10-00213-t001_Table 1Table 1Summary of in vitro and in vivo toxicity of GB.MaterialsSynthesized MethodAverage Lateral Dimensions (nm)In Vivo/In Vitro ModelsDosage (μg/mL)ResultsReferencesPGCVD50~1500Human dermal fibroblasts (HDFs) and L-929 fibroblast cells0~500Graphene showed a dose-dependent toxicity, and viability of HDFs were 80% with exposure of 15.6 μg/mL of Graphene for 24 h. No significant cytotoxicity of graphene towards L-929 was observed even at 200 μg/mL[83]PGCVDN.A.L929 fibroblasts cellsN.A.Compared to control group, 113.5% of cell proliferation was found on graphene coated glass, demonstrating that graphene not only is non-toxic to L929, but also promotes its cell proliferation. [84]Nitrogen doped grapheneElectrochemical exfoliation of graphite rods~1000Endothelial cells (HUVEC) and tumor–colorectal adenocarcinoma cells (DLD1)0~250Exposure of HUVEC cells to N-graphene with 2.33 wt% and 2.56 wt% caused decrease of cell viability; N-graphene has less impact on DLD1 cell viability.[98]GOHummer’s method~500Hela cells0~100Cell viability was higher than 80% and cells maintained a normal morphology after 48 h cocultured with 100 μg/mL GO. [85]GON.A.~1550 nm for mono-layered GO and ~120 nm for multi-layered (15~20) GODendritic cell (DC2.4)0~100Both GOs induced ROS and mono-layered GO has less effects on cell viability but caused a significant change in cell morphology.[88]Multi-layered GOHummer’s methodN.A. Human Erythrocytes and Skin Fibroblasts cells0~200Smaller-size GO showed lower hemolytic activity compared to larger-size GO.GO showed dose-dependent effect on viability of human skin fibroblasts cells. Viability was about 80% after 24 h exposure to 200 μg/mL GO. [89]Mono-layered rGOHydrothermal reductionN.A.Human Erythrocytes and Skin Fibroblasts cells0~200rGO showed low hemolytic activity due to low oxygen contents and aqueous aggregations.rGO showed dose-dependent effect on viability of human skin fibroblasts cells. Viability was about 80% after 24 h exposure to 12.5 μg/mL rGO. Its high toxicity was caused by aggregation.[89]rGOHydrazine hydrate reductionN.A.Human liver cells (HepG2)0~300rGO did not induce toxicity to HepG2 cells up to concentration of 50 μg/mL [99]rGOHydrazine hydrate reduction~500Human lung epithelial carcinoma cells (A549)0~400Hydrazine hydrate reduced GO exhibited a dose-dependent toxic effects, with a maximum of 70% cytotoxicity detected at 400 μg/mL.[87]rGOAscorbic acid reduction~500Human lung epithelial carcinoma cells (A549)0~400Increased protein leakage and disruption of membrane were observed in ascorbic acid reduced GO due to sharp protrusions caused during reduction.[87]

## 3. Three-Dimensional-Printed GBM-Based Nerve Guidance Conduits

### 3.1. Nerve Guidance Conduits

NGCs are tubular structures to enclose, or entubulate, the distal and proximal ends of the severed nerve, which can support and guide the regeneration of damaged peripheral nerves [100]. They bridge the gap between the severed nerve ends, providing a physical pathway and favorable microenvironment for nerve fibers to regrow and reconnect [101]. NGCs offer distinct advantages over traditional autologous nerve grafts for peripheral nerve repair. Unlike nerve grafts, NGCs are not limited by donor site availability, which can often be a constraint in reconstructive surgeries. Additionally, NGCs reduce the risk of surgical complications associated with donor site morbidity. Notably, NGCs demonstrate the capability to bridge larger nerve gaps exceeding 5 cm [2], expanding the potential applications in cases of extensive nerve injuries. As previously mentioned, potential materials for NGCs should be mechanically strong to provide structural support for regenerating tissues, electrically conductive to facilitate neuron transmission and signal propagation, porous to allow oxygen and nutrient exchange, and biocompatible and biodegradable to prevent inflammation and the need for a second removal operation. A number of materials have been explored for NGCs fabrication, including natural polymers (collagen [102], chitosan [27], silk fibroin [103]) and synthetic polymers (poly(lactic-co-glycolic acid) (PLGA) [104], PCL [105]). The U.S. Food and Drug Administration (FDA) has granted approval to several NGCs fabricated from collagen, poly(lactic-co-glycolic acid) (PLGA), or polyglycolic acid (PGA) [106]. Notably, the most recent FDA-approved NGCs include Axoguard Nerve Protector [107], derived from porcine submucosa extracellular matrix (ECM), in 2021. While allografted NGCs present a promising alternative, they often face challenges related to limited availability, variable quality, and potential immunogenicity, contingent upon donor and processing methodologies. Comparatively, NGCs fabricated from natural polymers may exhibit limitations in mechanical strength [108], degradation rate control [109], and potential for immunogenicity. Conversely, NGCs derived from synthetic polymers may raise concerns regarding biocompatibility and toxicity [110]. Since each material presents unique advantages and disadvantages and composite materials may overcome the limitations of individual materials.

### 3.2. Three-Dimensional Bioprinting Technology

Three-dimensional printing, also known as additive manufacturing, is an efficient and versatile method that produces 3D structures with high shape fidelity. This technology was first developed by Chuck Hull in the 1980s when he used ultraviolet light to harden photopolymers layer by layer to create a 3D structure [111], a technique known as stereolithography (SLA), one of the earliest additive manufacturing techniques. Today, 3D bioprinting methods include extrusion-based, inkjet-based, and light-based techniques, as illustrated in Figure 2, offering spatial resolutions from the nanoscale to the milliscale [112]. Additionally, hybrid approaches, such as melt electrowriting (MEW), combine additive and subtractive manufacturing methods to improve product quality [113]. The wide variety of 3D printing techniques enables the fabrication of diverse materials (from polymers to metals), sizes (from nanoscale to large components), and functionalities (ranging from self-assembling to heat transfer).

#### 3.2.1. Extrusion-Based 3D Bioprinting

Extrusion-based 3D bioprinting can be categorized into two primary sub-groups: processes based on material melting (fused deposition modeling, FDM) and processes without material melting (direct ink writing, DIW) [115]. In extrusion-based 3D bioprinting, bioinks—comprising biological materials with or without cells—are loaded into the print head and extruded through the nozzle layer by layer to create the desired structures. Consequently, bioinks used in extrusion-based printing must have specific properties, such as suitable viscosity, to ensure smooth extrusion through the nozzle and subsequent solidification on the printing bed. Depending on the process control, the printing resolution of this technique ranges from 100 μm to 1 mm [116]. Despite its limitation of relatively low resolution, this technique is widely employed in biomedical fields due to its broad compatibility with various materials, ease of handling, moderate environmental requirements, and cost-effectiveness.

#### 3.2.2. Inkjet-Based 3D Bioprinting

Inkjet bioprinting ejects ink droplets onto the substrates in a noncontact manner, making it suitable for high-viscosity bioinks while minimizing cell damage. This method offers higher cell viability (>85%) compared to extrusion-based bioprinting [117], making it advantageous for various bioapplications. Depending on the nozzle size and fluid properties, droplet sizes can range from 5 μm to 2000 μm [118]. As shown in Figure 3, inkjet printing can be classified into three types: continuous, drop-on-demand (DoD), and electrohydrodynamic (EHD) inkjet printing.

In continuous inkjet printing, ink is expelled from the nozzle under controlled pressure, forming a continuous stream that breaks into individual droplets due to Rayleigh-Plateau instability [119]. This method employs a droplet recovery device to recycle neutral droplets, reducing material waste. However, this process can result in ink contamination, making it unsuitable for biomedical applications. In contrast, DoD inkjet printing, which includes thermal, piezoelectric, and electrostatic jet printing, ejects droplets in response to specific signals, such as bubble generation or chamber deformation. DoD printing systems are simpler than continuous jet printing systems, as they do not require ink recycling, making them suitable for bioapplications. Additionally, the printing process can be precisely controlled by actuators, and DoD jet printers can be equipped with multiple print heads for multi-component material printing [120]. This technique offers high accuracy and efficient ink utilization. EHD jet printing, on the other hand, produces droplets through an electric field rather than relying on thermal energy or chamber deformation, making it particularly suitable for biomedical applications [121]. During EHD jet bioprinting, droplets are ejected when a specific voltage is applied between the nozzle and the substrate. EHD can generate droplets smaller than the nozzle size due to the ‘Taylor Cone’ effect [122], where the bioink forms a conical meniscus at the nozzle orifice. This technique offers higher resolution and improved printing capabilities, enabling the production of 3D structures with a wide range of materials, including hydrogels, cells, and proteins.

#### 3.2.3. Light-Based 3D Bioprinting

Light-based 3D bioprinting techniques utilize various light sources to polymerize bioinks before, during, or after extrusion. This approach offers a wide range of resolutions and construct sizes, from nanometers to centimeters, and supports multicellular and multimaterial bioinks. Common techniques include laser-based stereolithography (SLA), digital light processing (DLP), and two-photon stereolithography (2P-SL) (Figure 4) [123].

Traditional SLA uses a scanning mirror to direct a focused single-photon laser for polymerization. While this technique delivers high selectivity and resolution, its point-by-point scanning process is time-intensive, and the laser can harm living components in the bioink. In contrast, DLP improves efficiency by using a digital photomask to project an entire layer at once, enhancing printing speed in the x-y plane without sacrificing resolution [124], as shown in Figure 4. DLP also does not require supporting materials, providing greater flexibility in structure creation. However, it is limited to thin constructs due to its slower, multi-step, layer-by-layer process and is inefficient for multimaterial printing due to the need for repeated washing steps between layers.

Unlike single-photon absorption methods such as SLA and DLP, 2P-SL utilizes two-photon nonlinear absorption to achieve printing resolutions as fine as sub-100 nm [125]. This mechanism enables the use of NIR femtosecond lasers, which are less damaging to living cells and allow for deeper tissue penetration. In 2P-SL, two photons can interact with molecules either simultaneously or sequentially [126]. During simultaneous absorption, a super transient (~10^−16^ s) virtual state [127] is formed after the first photon is absorbed, requiring the second photon to arrive within this time to complete the process. In sequential absorption, an intermediate energy state forms between the two photon absorptions. Once the energy from both photons is absorbed, photoinitiators are excited from the ground state to a singlet or triplet state, triggering polymerization within the focused volume. This technique eliminates the need for a layer-by-layer approach, enabling the creation of truly 3D arbitrary structures. Despite its high resolution and low cell toxicity, 2P-SL is limited to smaller constructs and requires costly equipment.

#### 3.2.4. Combination of 3D Printing and Electrospinning Techniques

There are various hybrid manufacturing techniques that combine the strengths of different methods [128]. One prominent approach that has significant research attention is the combination of 3D printing and electrospinning for producing NGCs [129,130]. This hybrid method addresses limitations in both techniques: 3D printing’s relatively low resolution and electrospinning’s uncontrollable shapes. While 3D printing can fabricate porous structures, the pore sizes are often much larger than individual cells, which can negatively impact cell proliferation, migration, and differentiation [131]. In contrast, electrospinning uses high voltage to generate fibers with nanoscale diameters and well-defined tubular structures, closely mimicking the extracellular matrix (ECM) [132]. However, traditional electrospinning alone struggles to create fully 3D structures. By integrating these methods, it becomes possible to produce custom-shaped scaffolds with nano- or micro-scale interconnected pore networks. These nano- and micro-patterns, generated by electrospinning, can influence cell morphology and guide cellular arrangement. There are several approaches to integrating 3D printing and electrospinning. The simplest method involves first using a 3D printer to create a structure, onto which nanofibers are then deposited through electrospinning [133,134]. Alternatively, electrospinning can be used to fabricate a flat mesh or tubular nanofiber structure, with a 3D printer subsequently adding materials to reinforce or modify the scaffold [135]. A more advanced technique alternates between 3D printing and electrospinning, producing a hybrid structure composed of alternating layers of polymeric nanofibers and 3D-printed materials [136]. Additionally, electrospun fibers can be integrated directly into bioink, allowing for 3D-printed structures with a uniform fiber distribution, enhanced mechanical strength, and improved bonding between the fibers and the scaffold.

### 3.3. Design Strategies of 3D-Printed NGCs

Research has demonstrated that the microenvironment created by 3D structures is more effective than conventional 2D structures for the differentiation of stem cells [137]. This finding underscores the benefits of utilizing 3D printing techniques to fabricate NGCs. In a 3D structure, cells can be maintained in a controllable spheroid form, enhancing cell interaction and providing a superior niche for cell differentiation [138]. Despite potential concerns regarding cell viability due to the stress induced by printing pressure [139], 3D printing remains an attractive method for increasing cell homing when transplanting cells for targeted regeneration. Additionally, GBMs significantly enhance the mechanical strength and slow down the degradation rate of natural polymers, directly influencing axonal elongation and neuronal maturation [140]. Given these benefits, various design strategies have been developed to optimize 3D-printed NGCs, including material composition, porosity, and patterning techniques. These strategies aim to fine-tune the mechanical and biological properties of the conduits to enhance cell viability, guide nerve growth, and ultimately support functional recovery. The selection of a 3D printing method is closely tied to material properties, particularly the polymer matrix, and the intended structural characteristics of NGCs. As discussed in Section 3.2, extrusion-based techniques like DIW are suitable for high-viscosity bioinks, while inkjet-based printing requires low-viscosity inks. Light-based methods depend on photocrosslinkable materials, whereas hybrid approaches that integrate 3D printing with other fabrication techniques allow for the creation of more complex structures. The following sections will examine key examples and strategies for optimizing NGC designs using 3D-printed GBMs with various printing techniques. The selection of a 3D printing method is closely tied to material properties, particularly the polymer matrix, and the intended structural characteristics of NGCs. As discussed in Section 3.2, extrusion-based techniques like DIW are suitable for high-viscosity bioinks, while inkjet-based printing requires low-viscosity inks. Light-based methods depend on photocrosslinkable materials, whereas hybrid approaches that integrate 3D printing with other fabrication techniques allow for the creation of more complex structures. The following sections will examine key examples and strategies for optimizing NGC designs using 3D-printed GBMs with various printing techniques.

#### 3.3.1. Material Composition and Hybrid Structures

Proper materials design is crucial to achieve optimal properties of NGCs for peripheral nerve repair. One of the challenges of GBMs for NGCs fabrication is their poor dispersion, which can limit its effectiveness and biocompatibility. To overcome this, GBMs are often functionalized with polymers, improving their integration into NGC structures. Even small amounts of natural or synthetic biopolymers significantly enhance graphene dispersion. In turn, GBMs enhance the mechanical properties and electrical conductivity of polymer matrices, particularly those based on natural polymers. For instance, Song et al. [141] employed stable mixtures of graphene, gelatin, and sodium alginate (SA) to produce a shear shinning printable bioink, which enabled the fabrication of 3D scaffolds for NGCs using an extrusion-based 3D bioprinter. These graphene-enhanced scaffolds exhibit an increased compression modulus, greater hydrophobicity, decreased degradation rate, and enhanced PC12 cell proliferation and distribution. Another effective approach involves conjugating polyethylene glycol (PEG) with GBMs to improve their homogeneity and compatibility within the polymer matrix. This results in enhanced mechanical properties and electrical conductivity, which supports a favorable microenvironment for nerve regeneration. For example, Farzan et al. [142] developed a PEGylated GO and polyurethane (PU) resin bioink for fabricating grooved, hollow, or porous NGC tubes using SLA 3D printer, utilizing the photocrosslinkable properties of PU (Figure 5). PEGylation improved GO dispersion within the PU matrix, enhancing the wettability and hydrophilicity of the printed structure, which was favorable for cell attachment. The contact angle was decreased from 103° to 75° of PEG-GO, showing better cell attachment and cell proliferation. The PU/PEGGO NGCs exhibited controlled biodegradation, with 96% of PU/PEG-GO degraded by the end of week 6, maintaining structural integrity throughout the nerve healing process (approximately 6~12 weeks [143]).

Additionally, incorporating drugs or growth factors into GBMs/polymer composites has been explored to mitigate post-implant inflammation and promote nerve regeneration. Zhang et al. [144] integrated PCL and melatonin (Mel), the latter of which exhibits anti-inflammatory properties and promotes neural cell proliferation. Initially, GO hybrid SA fibers were printed by an FDM printer. Subsequent hydrothermal reduction of GO to rGO was performed at 90 °C for 24 h, with SA acting as the sacrificial component. The hollow rGO aerogel fiber was then coated in situ with PCL and Mel, resulting in rGO/PCL/Mel porous fibers with a porosity of up to 98.5% and an elastic modulus of 26.58 ± 4.99 MPa. In rat models with 15 mm nerve defects, muscle fiber area following transplantation reached 73.65% ± 3.81%, and the relative wet weight ratio of the target muscles was 0.59 ± 0.02, comparable to autograft controls. Similar to the incorporation of drugs and growth factors, seeding stem cells into NGCs would enhance the regeneration of nerves after implantation. Studies [138,145,146] have shown that MSCs differentiate more effectively into nerve cells in 3D-printed graphene-based hydrogels compared to a 2D environment. Additionally, Zhou et al. [147] used an extrusion-based 3D printer to fabricate rGO/methacrylate anhydride gelatin (GelMA) scaffolds, which were then seeded with SCs and MSCs. Immunohistochemistry and immunofluorescence analysis showed high expression of osteogenic and neural proteins in the rGO/GelMA scaffold, suggesting its potential to promote the synergistic regeneration of both nerve and bone.

#### 3.3.2. Structural Design for Cell Growth and Guidance

NGCs can be structured as single-channel porous or nonporous hollow tubes, with or without fillers, or as multichannel tubes that replicate the natural compartmentalization of nerves. The first NGCs were developed in 1985 [148] using nonporous poly(dimethyl siloxane) (PDMS) tubes, which effectively blocked scar tissue ingrowth but lacked permeability for small molecule exchange. Recent studies highlight the significance of oxygen and nutrient permeability in nerve regeneration. A pore size between 5 and 30 μm, with an optimal range of 10–20 μm, facilitates nutrient and waste exchange, endothelial cell migration, and prevents the outflow of growth factors and fibroblast infiltration [149,150,151,152]. One technique to control pore size involves using printing substrates, such as a rolling tubular model embedded with evenly distributed microneedles. For instance, Zhang et al. [153] integrated 3D printing and layer-by-layer casting methods to create NGCs microscale pores using a rolling tubular model with 50 μm microneedles as a printing substrate (Figure 6a). The 3D printer consisted of a rotating tube and a sprayer. A tubular mold with evenly distributed microneedles was placed on the roller, while the sprayer applied different solutions in successive layers. After printing, the microneedles were removed, leaving behind uniformly distributed micropores. During printing, a polydopamine (PDA) and arginylglycylaspartic acid (RGD) mixture was applied as the inner layer, followed by a graphene/PCL composite crosslinked with the PDA/RGD layer, resulting in an elastic modulus of 58.63 MPa. The outer layer of PDA/RGD was added to promote cell adhesion and proliferation. This approach created a uniformly porous tubular structure. Similarly, Fan et al. [154] utilized a 3D printer to create aligned pores on tubular PCL-GO NGCs produced through injection molding. Rather than printing 3D structures, the printer was employed specifically for micropore formation, with pore size controlled by the nozzle diameter. Initially, GO nanoparticles were mixed with PCL in dichloromethane (DCM) solutions, which were injected onto a three-layered tubular mold to form films as the DCM evaporated. A 3D printer was then used to create aligned pores on the NGC surfaces. These NGCs featured pores several micrometers in size, facilitating biofluid and nutrient exchange while being small enough to prevent the entry of alien cells such as fibroblasts, yet allowing essential elements to pass through.

Combining electrospinning with 3D printing allows the fabrication of multi-layer NGCs. Sun et al. [155] developed multiscale NGCs with PCL microfibers, rGO/PCL microfibers, and PCL/collagen nanofibers as multiscale hollow NGCs (MH-NGCs), while PCL and rGO/PCL microfibers were used as a filler (Figure 6c). Firstly, 10 μm PCL microfibers and 125 μm rGO/PCL microfibers were printed sequentially by MEW, providing anisotropic guidance for neuronal growth and mechanical stability of the structure. Then, PCL/collagen nanofibers were electrospun to form the outer layer of the MH-NGCs. Next, PCL and rGO/PCL microfibers were MEW-printed, rolled into a filler, and packed into the MH-NGCs to produce multiscale filled NGCs (MF-NGCs). This combination of micro- and nanofibers resulted in a biomimetic structure designed to support nerve regeneration. These MF-NGCs demonstrated excellent mechanical stability, retaining 96% of their compressive strength after 100 cycles, and significantly improved conductivity due to rGO incorporation. In vitro studies showed over 95% cell viability for RSC96 and PC12 cells after seven days, with PC12 cells displaying elongated growth along the aligned microfibers. Animal models further confirmed successful nerve regeneration in a 10 mm peripheral nerve defect using these MF-NGCs.

The structural design of NGCs is critical for enhancing cell adhesion, proliferation, and differentiation. For instance, multichannel tubular designs, while more complex to fabricate, offer superior biological integration. Researchers have used EHD jet printing to create rGO NGCs with synthetic and natural polymers [156,157,158]. EHD printing allows for the incorporation of active ingredients in a single step, producing anatomically precise nerve grafts. Figure 7 illustrates the fabrication of graphene-loaded PCL/PEO dual-core matrices, where gelatin and dopamine hydrochloride (DAH) were encapsulated within PEO cores. DAH enhances biocompatibility [159], where gelatin and dopamine hydrochloride (DAH) were encapsulated within PEO cores. DAH enhances biocompatibility [130], while gelatin improves hydrophilicity and cellular affinity [160]. These dual-core NGCs exhibited diffusion-based DAH release, enhanced biocompatibility, and greater cell viability with higher graphene concentrations compared to graphene-free matrices.

In addition to porosity and multichannel designs, studies have shown that nerve guidance conduits (NGCs) with highly aligned topographical and micropatterns effectively guide axonal growth along the conduit structure. These topographical cues lead to a mature neuronal phenotype without the need for exogenous growth factors or stimulants [161,162,163]. For instance, Filippini et al. [164] utilized an FDM printer with graphene-PLA filaments to print scaffolds featuring 100 μm and 400 μm spaces. Results showed that scaffolds with 100 μm spacing promote the alignment of myoblast and fibroblast cells and guide neurites along the scaffold’s line pattern. Additionally, Chen et al. [165] employed MEW to fabricate anisotropic, microfibrous PCL architectures functionalized with GO and graphitic carbon nitride (g-C_3_N_4_) (Figure 8). The 10-layer grid-patterned structures (30 mm × 30 mm) with varying fiber spacings (200, 400, and 600 μm) guided neurite growth along the fibers under photocatalytic stimulation by GO and g-C3N4. Similarly, another study [140] utilized an FDM printer to extrude a PCL/GO composite onto silicon substrates with groove patterns (1 μm, 5 μm, and 10 μm wide, with nanoscale heights of 80 nm, 210 nm, and 280 nm) to create stamps. These patterned stamps were then rolled into tubes, forming NGCs. Compared to soft lithography, the 3D printing approach provided similar resolution but resulted in more reproducible stamps with fewer defects. The NGCs with micro/nano topographies facilitated a balanced response in cell adhesion, alignment, and neurite extension, thereby guiding cell migration. Given that neuronal cell bodies typically range from 10 to 50 μm, and radial glial cells form fibers approximately 1 μm in diameter, the study revealed that PCL/GO patterns with a 10 μm width and a 280 nm height allowed for enhanced neurite branching, potentially promoting greater neuronal differentiation.

#### 3.3.3. Electrical Conductivity for Enhanced Nerve Regeneration

GBMs’s superior conductivity plays a key role in enhancing the transdifferentiation of mesenchymal stem cells (MSCs) into Schwann cells (SCs) through electrical stimulation, a critical aspect of peripheral nerve therapies [166]. The electrical conductivity required for nerve cells to communicate effectively depends on their ability to transmit action potentials, which are rapid changes in voltage across the cell membrane. In neurons, the conductivity is facilitated by ion channels, where the flow of positive sodium ions generates electrical signals [167]. Neurons rely on a voltage difference for rapid transmission of electrical impulses, which is vital for neuronal communication. The typical electrical conductivity of human nerve is 0.08–1.3 S/m [168], meaning NGCs must meet or exceed this range to effectively support neuronal stimulation and regeneration. GBMs can effectively enhance cell differentiation by improving the conductivity of NGCs, allowing better propagation of electrical signals, especially in nerve regeneration applications (Table 2). Researchers [145] used an FDM printer to fabricate a graphene-based interdigitated circuit from conductive graphene PLA filament (Figure 9). This circuit was embedded into a gelatin solution to create a graphene-based gelatin 3D porous scaffold designed to induce MSC differentiation. The scaffold exhibited a resistance of approximately 50 kΩ, sufficient to power an LED and deliver the electrical stimuli necessary for promoting cell differentiation. With 90% porosity and an average pore size of ~135 μm, the 3D-printed scaffold closely mimicked the extracellular matrix (ECM) microenvironment, achieving an 80% transdifferentiation rate of MSCs into SCs.

This study demonstrated that poly(lactic acid)/poly(ε-caprolactone)/graphene (PLA-PCL-G) NGCs printed by an FDM printer with 1.5 wt% of graphene have a conductivity of $8.2\times 10^{-3}\;S/m$, compared to the nonconductive nature of PLA and PCL. However, this electrical conductivity is still lower than that of human nerves [169]. Covalently bonding GBMs with polymers can effectively enhance the electrical conductivity of printed composites [170,171]. Park et al. [170] fabricated a double methacrylated gelatin (GelMA) to covalently bond with the carboxyl groups on GO through a carbodiimide reaction. A bioink composed of silk fibroin and GO/GelMA was then printed using a DLP 3D printing technique, demonstrating an electrical conductivity of 15 S/m due to the zero-length crosslinking method of GO/GelMA.

Increasing the amount of GBM can also significantly enhance the electrical conductivity of 3D-printed scaffolds, which in turn promotes the differentiation of stem cells into neuron-like cells. Shah et al. [172] used an extrusion-based DIW printing method to develop 3D scaffolds containing graphene and polylactide-co-glycolide (PLG). During extrusion, shear forces facilitate the reorientation and alignment of graphene flakes along the flow direction through a 100 µm nozzle, resulting in 3D scaffolds with anisotropic microstructures and properties (Figure 10). With 20% graphene content, the scaffold achieved a conductivity of 50 S/m, which increased to 800 S/m with 60% graphene. Morphologically, hMSCs on 20% graphene exhibited a confluent, sheet-like structure typical of adherent cells such as fibroblasts. In contrast, on 60% graphene, hMSCs developed highly elongated structures, forming high aspect ratio cellular extensions that created “wire-like” networks. By day 14, some cells displayed morphologies similar to uni- or multipolar neurons, including approximately 2 μm diameter axon-like extensions and features resembling presynaptic terminals.
biomimetics-10-00213-t002_Table 2Table 2Electrical properties of 3D-printed GBM-based NGCs.Materials of NGCsFabrication TechniqueElectrical Conductivity (S/m)Cell Behaviors and Axon Formation ResultsReference5% PEGylated GO-PUSLA1.1×10−3N.A.[142]Porous rGO/PCL/Mel fiberDIW0.109Axon formation was observed in a Sprague–Dawley rat model, with an average diameter of 4.98 ± 2.24 μm and a thickness of 0.59 ± 0.36 μm, comparable to that of autografted NGCs.[144]PDA/RGD-SG/PCLHybrid technique (casting and 3D printing)0.892In vitro results showed that SC neural expressions were improved, and in vivo results exhibited functional sciatic nerve recovery and axon regrowth.[153]PDA/RGD-MG/PCLHybrid technique (casting and 3D printing)0.637In vitro results showed that SC neural expressions were higher than PDA/RGD/PCL but slightly lower than PDA/RGD-SG/PCL, and in vivo results exhibited functional sciatic nerve recovery and axon regrowth.[153]GO/PCLHybrid technique (injection molding and 3D printing)4.55×10−4In vivo results showed that GO/PCL NGCs exhibited a significantly greater total number, area, diameter, and thickness of regenerated nerves and myelinated axons compared to the PCL group, similar to autograft NGCs at 18 weeks post-surgery.[154]rGO/PCLEHD0.135The addition of rGO results in softer scaffolds support neural differentiation of PC 12 cells [157]rGO/Silk FibroinDLP6.5In vitro results showed the printed NGCs promote the proliferation of Neuro2a cells and exhibit neurogenic activity by inducing neuronal differentiation in neuroblastoma cells.[171]GO/GelMADLP15Neuro2a cells exhibit more pronounced neurite growth when encapsulated in GO/GelMA compared to when treated with differentiation induction media.[170]Graphene-PLA-PCLFDM8.2×10−3N.A.[169]20% Graphene-PLGDIW50Human MSCs (hMSCs) exhibited a confluent, sheet-like morphology, indicating characteristic of adherent cell types, such as fibroblasts[172]60% Graphene-PLGDIW800Human MSCs exhibited axon-like extensions and features on day 14[172]

## 4. Conclusions and Future Perspective

In recent years, the integration of 3D printing technology with GBMs has greatly promoted the development of NGCs. This innovative production method allows NGC to be customized and personalized, taking advantage of the superior properties of GBM, such as mechanical strength, electrical conductivity, and biocompatibility, which promotes cell embedding, differentiation, and proliferation within NGCs.

This review highlights the significant benefits of integrating GBM into 3D-printed NGCs. GBM’s superior mechanical strength and conductivity address the critical dual challenge of supporting cellular function while ensuring structural integrity. In addition, the versatile functionalization of GBMs enhances their interaction with biological tissues, making them ideal for neurological applications. Despite these advantages, there are several challenges that hinder the widespread clinical application of GBM-based NGCs.

Firstly, the long-term biocompatibility and cytotoxicity of GBM-based NGCs require comprehensive investigation. Although the toxicity of GBMs can be tailored through factors such as dosage, dimensions, and functionalization, the long-term effects of these materials need careful consideration, especially when integrated with degradable materials in NGCs. Evaluating the biological response to materials released upon degradation is crucial. Additionally, achieving uniform dispersion of GBMs within polymer matrices as bioinks remains a significant technological issue. Aggregation not only affects mechanical and electrical properties but also raises biocompatibility concerns.

Secondly, despite the rapid advancements and increased maturity and intelligence of 3D printing technology, with some standards already established, a fully robust system has yet to be achieved. Attaining high resolution and shape fidelity in NGCs continues to be challenging, especially when developing the sub-micro and nanoscale patterns critical for optimal nerve regeneration. This emphasizes the necessity for further exploration into hybrid or combined fabrication techniques.

The existence of these problems limits the comprehensive promotion of GBM-based NGCs in clinical applications. Addressing these challenges through systematic studies and technological innovation will be essential to unlocking the full potential of graphene-based materials in nerve repair applications.

For the future, the combination of 3D printing and GBMs is expected to achieve wider applications in the field of neuroprosthetics and regenerative medicine. To solve the current challenges, the following should be paid tremendous attention:

Biocompatibility and cytotoxicity: The biocompatibility and cytotoxicity of GBMs should be well and deeply explored via systematic study. In addition, novel composites based on GBMs should be designed and developed to improve biocompatibility and cytotoxicity.

Advanced manufacturing techniques: The printing accuracy and functionality of NGCs based on GBMs should be improved via combining 3D printing and other manufacturing methods.

Clinical validation: The clinical safety of GBM-based NGCs should also be systematically evaluated via strengthening collaboration with clinical practitioners to conduct multicenter clinical trials, which would promote its clinical application.

## Figures and Tables

**Figure 1 biomimetics-10-00213-f001:**
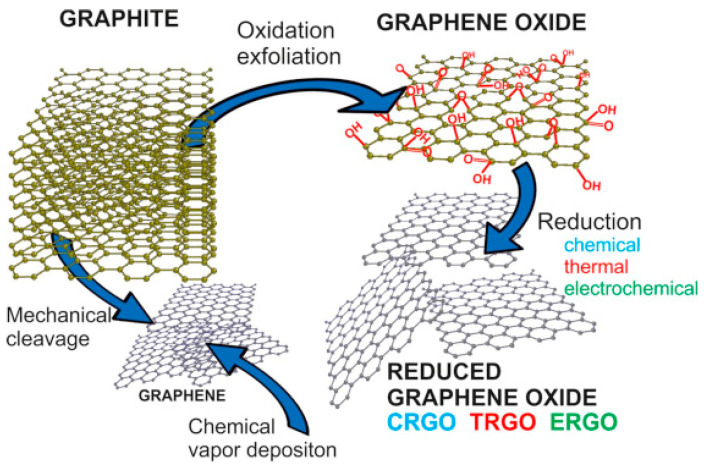
Synthesis methods of PG, GO, and rGO [45].

**Figure 2 biomimetics-10-00213-f002:**
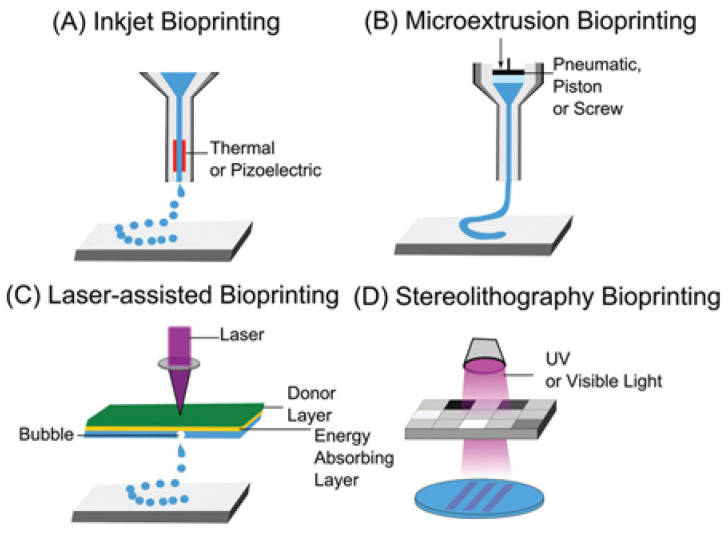
Examples of different bioprinting techniques [114].

**Figure 3 biomimetics-10-00213-f003:**
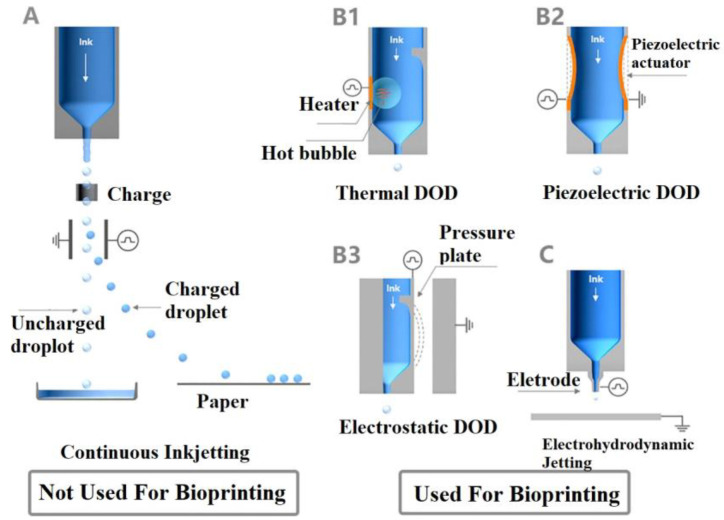
Schematic demonstration of inkjet printing methods: (**A**) *Continuous inkjet printing*: Liquid is dispersed into continuous droplets, with droplet designation controlled by electric charge and field. (**B**) *Drop-on-demand (DOD) inkjet printing*: Droplets are ejected selectively to form patterns. (**B1**) *Thermal inkjet*: Droplets are ejected by bubble formation through heating. (**B2**) *Piezoelectric inkjet*: Droplets are generated via vibration of a piezoelectric actuator. (**B3**) *Electrostatic inkjet*: Droplets are ejected by deforming a pressure plate. (**C**) *Electrohydrodynamic jet printing*: Droplets are produced using a high-voltage electric field [118].

**Figure 4 biomimetics-10-00213-f004:**
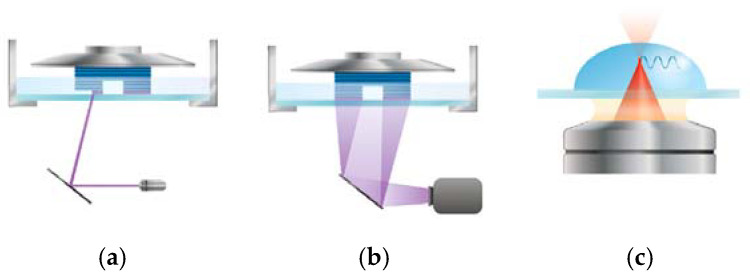
Schematic demonstration of (**a**) SLA, (**b**) DLP, and (**c**) 2P-SL printing techniques [123].

**Figure 5 biomimetics-10-00213-f005:**
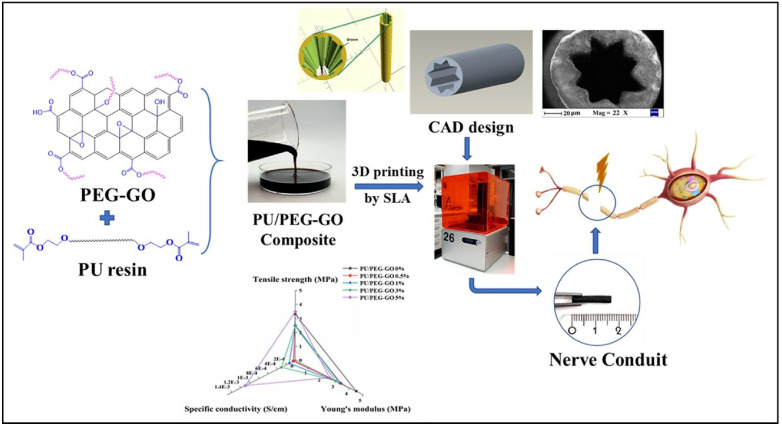
Flow chart of 3D-printed PU/PEGGO for NGCs [142].

**Figure 6 biomimetics-10-00213-f006:**
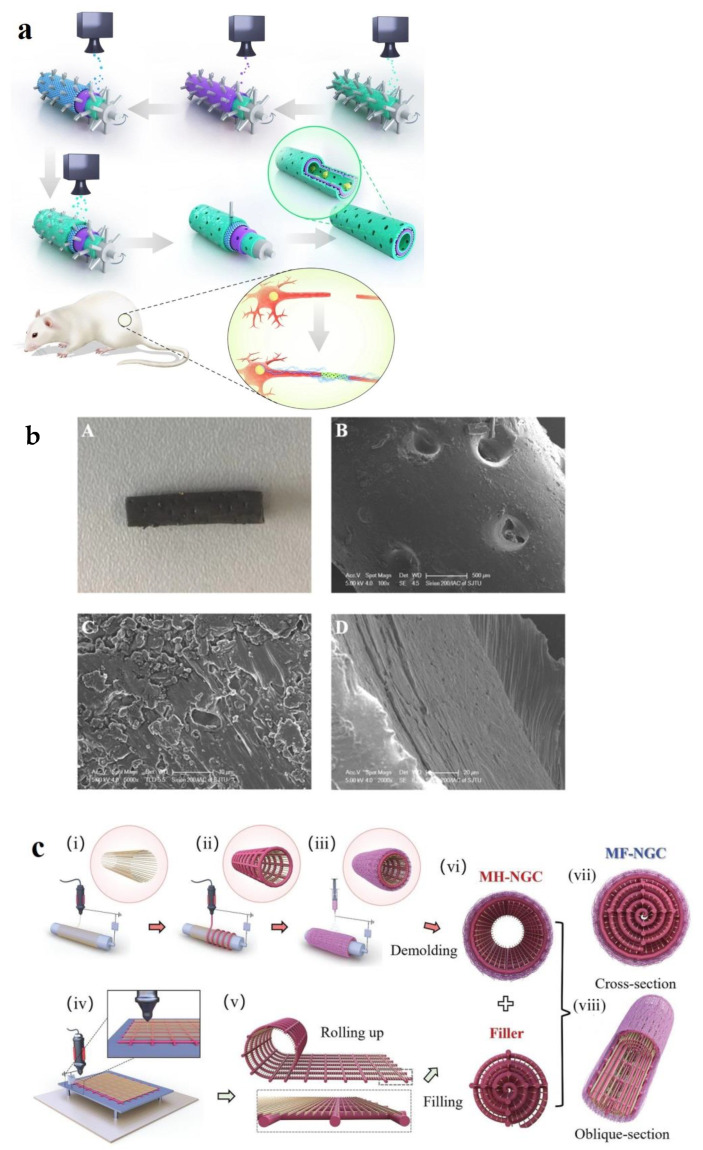
(**a**) Schematic illustration of GBM-based NGCs fabricated using a rolling tubular model with 50 μm microneedles. The green layers are PDA/RGD mixed layers, the purple layer is graphene and PCL mixed layer and the blue layer is a repetition of the graphene and PCL mixed layer [153]. (**b**) (**A**): optical images of GO/PCL NGCs; (**B**,**C**): SEM images showing the nanoporous structure of the GO/PCL NGCs; (**D**): TEM images showing the uniform distribution of GO nanoparticles in PCL scaffolds [154]. (**c**) Schematic illustration of the fabrication of multiscale filled NGCs: (**i**) MEW printing of PCL microfibers on a rotating mandrel; (**ii**) MEW printing of rGO/PCL microfibers; (**iii**) electrospinning of PCL/collagen nanofibers and removal from the mandrel to obtain an MH-NGC; (**iv**) MEW printing of a fibrous sheet comprising PCL microfibers (yellow) and rGO/PCL microfibers (red); (**v**) rolling the fibrous sheet; and (**vi**) inserting the densely packed fibrous sheet into the lumen of a multiscale hollow NGC to create an MF-NGC. (**vii**) Schematic of 3D-printed MF-NGCs: (**vii**) cross-sectional view of the MF-NGC and (**viii**) oblique-sectional view of the MF-NGC [155].

**Figure 7 biomimetics-10-00213-f007:**
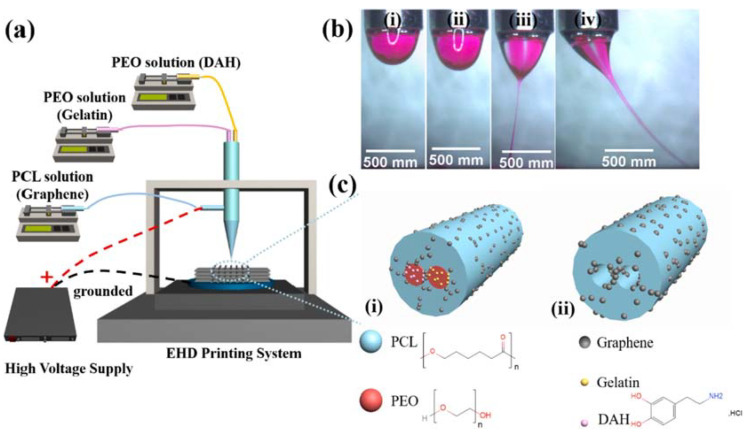
(**a**) Schematic of the dual-core EHD printing system. (**b**(**i**–**iv**)) Sequential images of dual-core jet formation during dual-core filament production. (**c**) Diagram of graphene-loaded dual-core fibers: (**i**) before and (**ii**) after the drug release process [156].

**Figure 8 biomimetics-10-00213-f008:**
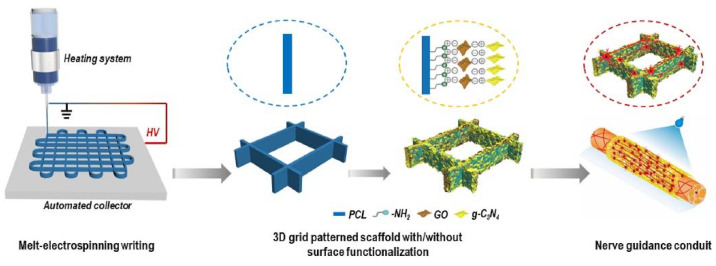
Schematic illustration of 3D polymeric grid-patterned scaffolds, decorated with a visible-light photocatalyst, serving as NGCs to enhance peripheral neural regeneration [165].

**Figure 9 biomimetics-10-00213-f009:**
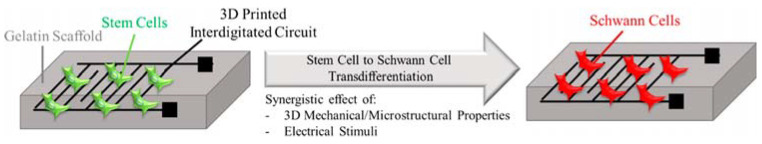
Schematic illustration of 3D-printed graphene-based interdigitated circuit activate MSCs differentiation [145].

**Figure 10 biomimetics-10-00213-f010:**
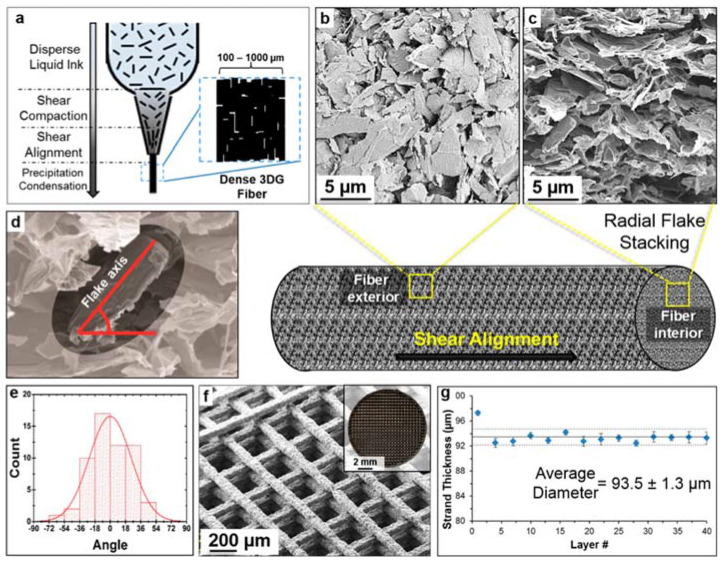
(**a**) Illustration of dispersed liquid inks prior to extrusion. (**b**,**c**) SEM images of the fiber exterior and cross-section, respectively. (**d**) Example illustrating the measurement of flake orientation in an end-on cross-sectional view of the 3DG fiber. (**e**) Histogram of graphene flake orientations with respect to the horizontal. (**f**) SEM and optical (inset) images of 3D scaffolds printed with a 100 μm tip. (**g**) Uniformity of 3D scaffolds quantified by fiber thickness in a 40-layer construct printed with a 100 μm tip [172].

## Data Availability

A preprint version of this manuscript was posted on Authorea (https://wileyopenresearch.authorea.com/doi/full/10.22541/au.173556927.78612466/v1) on 30 December 2024. This submission has been updated and revised since the initial preprint version.

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
