# Peer review of "A Comprehensive Review on Bioprinted Graphene-Based Material (GBM)-Enhanced Scaffolds for Nerve Guidance Conduits"

_biomimetics, 2025, doi:10.3390/biomimetics10040213_

Round 1
Reviewer 1 Report
Comments and Suggestions for Authors
Upon reviewing the manuscript entitled (Recent advances on bioprinted graphene-based materials (GBMs) enhanced scaffolds for nerve guidance conduit), it seems to be an interesting work.
The literature survey is adequate. However, the cited references are insufficient.
The text and language have to be revised. Some typo appear and repeated words as well.
Moreover, there are some comments require to be replied.
1- In the Introduction section, the authors have to focus on the novelty of this work when compared to previous studies.
2- In section (2.3.3 Cytotoxicity Properties), the text (While some research highlights the ability of GBMs to enhance cell growth, others raise concerns about their potential health risks.) is not clear. Please clarify in detail.
3- The text of Fig3 is illegible. Please provide a better figure.
4- I section (3.3.3 Electrical Conductivity for Enhanced Nerve Regeneration), (Therefore, electrical conductivity is essential in the design of NGCs to control cell growth, proliferation, migration, and differentiation.). What are the conditions of the required electrical conductivity? Please explain.
5- The Conclusion needs to be rewritten.
6- More recent and relevant references are recommended to be cited in this articles.
Comments on the Quality of English Language
The text and language have to be revised.
Some typo appear and repeated words as well.
Reviewer 2 Report
Comments and Suggestions for Authors
The manuscript titled "Recent advances on bioprinted graphene-based materials enhanced scaffolds for nerve guidance conduit" mostly reviews the mechanical and cytotoxic properties of bioprintable graphene-based scaffolds (GBMs), as well as some examples of their application for treating nerve injuries. The review also contains basic information on graphene, its structure, types, and synthesis methods, as well as bioprinting techniques. However, there is a slight discrepancy between the title, the goal of the article, which is stated in the introduction, and the content of the review. Specifically, it is mentioned that "In this review we focus on 3D bioprinted GBMs-enhanced scaffolds for nerve conduits summarizing recent advancements and future perspectives in this research area." However, in reality, not much attention is given to recent advances, and the review mainly provides basic information about GBMs for NGC and discusses the possibilities of their use. This discrepancy may lead to confusion and loss of interest among future readers.. May be, the introduction should be rewritten according to the review's text, and the title of the manuscript should also be rethought. Other flaws are listed below:
1) Lines 52-65 need references, especially points 3-5 need appropriate references for the demanded criteria for materials used in nerve injury treatment.
2) Lines 69-72 also require appropriate references to some papers where GBM properties were described and studied.
3) Section 3.2 describes existing methods of bioprinting, which may be applied to different materials. However, there is no information about what approaches are most often used or preferable for GBMs in subsequent sections. Also, when describing cited works, the method of bioprint used for each material is not mentioned in section 3.3, which makes it confusing to focus so much on bioprint methods in one section and not discuss the approaches used for cited materials in another.
4) The most important section of the review, Section 3.3, did not review enough recent developments in GBMs for NGCs. This section contains only 5 references from the years 2020 to 2024. While there are many published reviews, including those published in MDPI journals close to the present one, which considered many other papers. For example: https://doi.org/10.3390/biomedicines10010073 and DOI: 10.1371/journal.pone.0229863
Comments on the Quality of English Language
English language needs correction. Some grammar mistakes and typos were found in the following lines
-lines 42-44 may be preposition was missed. "... as gold standard ..." looks better
-line 262 However is written twice
-in vitro and in vivo should be written in italics
These are some several points, the manuscript need proper English language and style revision.
Round 2
Reviewer 1 Report
Comments and Suggestions for Authors
The revised version of this manuscript appears in a better structure than the previous one.
The text has been improved and refined. I can recommend accepting it for publication in the current form.
Reviewer 2 Report
Comments and Suggestions for Authors
All the comments were reviewed, the manuscript can be accepted as it is.